# Identifying the Compounds of the Metabolic Elicitors of *Pseudomonas fluorescens* N 21.4 Responsible for Their Ability to Induce Plant Resistance

**DOI:** 10.3390/plants9081020

**Published:** 2020-08-12

**Authors:** Helena Martin-Rivilla, F. Javier Gutierrez-Mañero, Ana Gradillas, Miguel O. P. Navarro, Galdino Andrade, José A. Lucas

**Affiliations:** 1Plant Physiology Pharmaceutical and Health Sciences Department, Faculty of Pharmacy, Universidad San Pablo-CEU Universities, 28668 Madrid, Spain; jgutierrez.fcex@ceu.es (F.J.G.-M.); alucgar@ceu.es (J.A.L.); 2Centre for Metabolomics and Bioanalyses, Faculty of Pharmacy, Universidad San Pablo-CEU Universities, 28668 Madrid, Spain; gradini@ceu.es; 3Laboratory of Microbial Ecology, Department of Microbiology, Londrina State University, Londrina 86051-990, Brazil; micromiguel@gmail.com (M.O.P.N.); andradeg@uel.br (G.A.)

**Keywords:** *Pseudomonas fluorescens* N 21.4, metabolic elicitors, isoflavone elicitation, induced systemic resistance, sphingolipids, terpenoids

## Abstract

In this work, the metabolic elicitors extracted from the beneficial rhizobacterium *Pseudomonas fluorescens* N 21.4 were sequentially fragmented by vacuum liquid chromatography to isolate, purify and identify the compounds responsible for the extraordinary capacities of this strain to induce systemic resistance and to elicit secondary defensive metabolism in diverse plant species. To check if the fractions sequentially obtained were able to increase the synthesis of isoflavones and if, therefore, they still maintained the eliciting capacity of the live strain, rapid and controlled experiments were done with soybean seeds. The optimal action concentration of the fractions was established and all of them elicited isoflavone secondary metabolism—the fractions that had been extracted with n-hexane being more effective. The purest fraction was the one with the highest eliciting capacity and was also tested in *Arabidopsis thaliana* seedlings to induce systemic resistance against the pathogen *Pseudomonas syringae* pv. tomato DC 3000. This fraction was then analyzed by UHPLC/ESI–QTOF–MS, and an alkaloid, two amino lipids, three arylalkylamines and a terpenoid were tentatively identified. These identified compounds could be part of commercial plant inoculants of biological and sustainable origin to be applied in crops, due to their potential to enhance the plant immune response and since many of them have putative antibiotic and/or antifungal potential.

## 1. Introduction

Protecting crops against diseases caused by pathogens in agricultural systems has always been a constant challenge when trying to maximize crop yields, minimize economic losses and ensure quality food worldwide [1]. As 2020 has been declared International Year of Plant Health by the FAO (Food and Agriculture Organization of the United Nations), the current challenge is to find effective, ecofriendly and at the same time, low-cost agriculture control methods that guarantee the sustainability of crop production while eliminating negative impact on the environment.

It has been widely demonstrated that biological agents, such as beneficial microorganisms, are able to enhance plants’ immune systems, inducing systemic resistance (ISR) and/or systemic acquired resistance (SAR) [2,3,4]. This phenomenon of enhancing a plant’s immune system is called elicitation, and it supposes that cells exposed to external factors activate defense mechanisms by triggering and regulating some biochemical and molecular responses, increasing the synthesis of specific molecules with a protective role [5]. After elicitor perception, signals are transported throughout the plant, triggering local and systemic responses [6] and leading to the generation of reactive oxygen species (ROS), phytoalexin biosynthesis, increased synthesis of antioxidant secondary metabolites, reinforcement of plant cell wall, deposition of callose, synthesis of defense enzymes, accumulation of pathogenesis-related proteins [7], etc.

Originally, the term elicitor referred to molecules capable of inducing the production of phytoalexins, but nowadays it is commonly used for compounds that stimulate any defensive line [8,9,10]. Natural elicitor molecules derived from microorganisms can induce similar defense responses in plants to those responses induced by the alive microorganisms [11]. Different types of natural elicitor molecules have been characterized, including structural molecules, such as carbohydrate polymers, lipids, and bacterial flagellin [12], and metabolic elicitors, released to the medium, such as antibiotics and secondary metabolites [13,14].

Currently, a wide range of metabolic elicitors produced by fungi and beneficial rhizobacteria have been found to induce in plants the synthesis of protective compounds—such as phytoalexins, defensins, phenolic acids, and flavonoids—that directly suppress pathogens [15]. They can reduce plant diseases through elicitation of physical and chemical processes linked to systemic plant defense mechanisms [16]. This is why plant inoculants made of beneficial rhizobacteria and/or their metabolic elicitors have been seen as feasible and effective alternatives to chemical phytosanitaries to counteract the attack of pathogens and also to face to diverse biotic and abiotic stresses [17,18]. However, using elicitor molecules instead of living bacteria is a way to reduce the cost and to simplify the production and subsequent management of plant inoculants. Products made of partially purified or purified compounds derived from bacterial metabolism [19] are cheap to produce; easy to manage; respectful with the environment; not harmful or toxic to human health or to other organisms, as their targets are directly the plants; and do not cause biosecurity problems as bacteria could cause [20]. Furthermore, they can be easily applied by spraying [21] and they do not require special preservation conditions like live microorganisms because they are stable during long periods of exposure to light and/or high temperatures [22] and they do not lose viability during prolonged storage.

Among beneficial microorganisms, the genus *Pseudomonas* spp. and *Bacillus* spp. are the most studied genera concerning everything related to plant–pathogen–beneficial microorganism interactions and the improvement of plant immune system, because they are very abundant in the soil and because of their role in the suppression of pathogens [23]. Literature has shown that *Pseudomonas* spp. have extensive metabolic capabilities and adaptable biochemistry through their production of structurally varied bioactive molecules [24]. Furthermore, it is largely known that *Pseudomonas* spp. are some of the most important microorganisms able to produce compounds with antibiotic or eliciting activity triggering SAR [25] or ISR [3].

The potential of *Pseudomonas* spp. to suppress plant pathogens has been demonstrated in many plant species and around the world [26]. They can be used as efficient and not-risky biocontrol agents to use in agriculture because they do not show pathogenic, allergenic or harmful risks to people or animals [27]. Secondary metabolites isolated from *Pseudomonas* spp. that could be an alternative to the use of chemical compounds in the control of plant disease include phenazines, pyrrolnitrin-type antibiotics, betalactones, pyo compounds, indol derivatives, peptides, glycolipids, lipids, aromatic organic compounds and aliphatic compounds, among others [24,28].

More specifically and within *Pseudomonas* spp. genus, the specie *Pseudomonas fluorescens* is a Gram-negative soil bacterium that has been widely studied in relation to its capacity to suppress other pathogenic microorganisms by producing siderophores, antibiotics and antifungal and antiparasitic compounds and to induce systemic resistance in plants through a vast variety of secondary metabolites with eliciting capacity [28,29,30,31]. For all of this, certain strains have already been developed as commercial products for management of plant illnesses in agricultural settings [22].

The *P. fluorescens* strain N 21.4 was specifically used in the present work because its capacity to induce systemic resistance in different plant species, such as *Arabidopsis thaliana* [18,32], *Solanum lycopersicum* [33], *Hypericum* sp. [34], *Papaver* sp. [35] and blackberry (*Rubus* cv. Loch Ness) has been largely demonstrated [36,37,38,39]. Its metabolic elicitors have also been demonstrated to induce systemic resistance in *A. thaliana* [18] and to elicit flavonoid metabolism in the leaves and in the fruits of cultivars of blackberry [38,39].

For all the above, the objective of the present study was to isolate, purify, test and identify the compound or set of elicitor compounds of *P. fluorescens* N 21.4, obtained from sequential fractionations of its metabolic elicitors by vacuum liquid chromatography (VLC), responsible for plant elicitation. To corroborate fractions’ capacities to elicit secondary metabolism, some experiments under controlled conditions were performed in soybean seeds to enhance isoflavone synthesis and in *A. thaliana* to induce systemic resistance. A final analysis to obtain the profile of main compounds present in the purest fraction of the metabolic elicitors was made by ultra-high performance liquid chromatography (UHPL), with an electrospray ionization source (ESI) and a quadrupole time-of-flight mass spectrometry analyzer (QTOF-MS). Eight compounds were tentatively identified and classified into different families: alkaloids, amino lipids, terpenoids and arylalkylamines.

## 2. Material and Methods

Sequential extraction, fragmentation and purification of the metabolic elicitors of *P. fluorescens* N 21.4 were carried out (Figure 1 and Table 1). Firstly, a liquid–liquid phase separation was made [40], followed by two sequential VLCs. All the sequentially fragmented fractions were inoculated in soybean seeds to analyze their capacity to elicit isoflavone secondary metabolism pathway. Final isolation and purification were performed. The resulting fraction of the entire process of fragmentation and purification was tested in an ISR experiment in *A. thaliana* seedlings. Finally, this fraction, the purest one, was analyzed by UHPLC/ESI–QTOF–MS to characterize its composition.

### 2.1. Bacterial Strain

The bacterial strain used in this study was *P. fluorescens* N 21.4 (Spanish Type Culture Collection accession number CECT 7620), a Gram-negative bacilli isolated from the rhizosphere of *Nicotiana glauca* Graham [33]. The bacterial strain was stored at −80 °C in nutrient broth (CONDA) with 20% glycerol.

### 2.2. Metabolic Elicitor Extraction and Control Obtaining

The bacterial strain, stored at −80 °C in nutrient broth with 20% glycerol, was streaked onto nutrient agar (peptone 3 g·L^−1^, beef extract 5 g·L^−1^ and agar 15 g·L^−1^ pH 7) plates and cultivated for 24 h at 28 °C. After 24 h of growth, bacterial cells were scraped off the plates into 10 L of sterile nutrient broth (peptone 3 g·L^−1^ and beef extract 5 g·L^−1^ pH 7) and incubated on a rotatory shaker at 28 °C and 180 rpm for 24 h.

Metabolic elicitors (released into the medium) were obtained by centrifuging the 10 L of N 21.4 culture at 2890× *g* during 20 min at 4 °C. Cells were discarded and the remaining supernatant was evaporated in a stove at 60 °C until obtaining 1 L. This concentrated supernatant was filtrated through a 0.2 µm nitrocellulose filter and extracted twice with a double volume of hexane (*v*/*v*). The extract was evaporated to dryness in a Buchi R-215 rotary evaporator at 50 °C [40]. The dry extract was weight (250 mg) and stored at 4 °C protected from light and humidity.

To obtain the control 1, the same procedure was followed as for extracting the metabolic elicitors from the bacterium, but while carrying out the entire process exclusively with the nutrient medium (peptone 3 g·L^−1^ and beef extract 5 g·L^−1^ pH 7), in the absence of bacterium.

### 2.3. Elicitation of Isoflavone Metabolism in the Soybean

To test the capacity of the sequentially obtained fractions to elicit isoflavone secondary metabolism, rapid induction tests were carried out in soybean seeds (BS-2606 Embrapa). Seeds were superficially disinfected with a 70% ethanol bath for 1 min, 5% sodium hypochlorite for 6 min and 5 washes with sterile distilled water. After that, seeds were kept imbibing in sterile distilled water for 4 h, in darkness and at room temperature. After imbibition, 90 seeds per treatment and 90 seeds for each control (control 1 and 2), were distributed in 3 replicates of 30 seeds each and put to germinate in sterile Petri dishes with 1% European bacteriological agar. A small longitudinal cut was made in the seeds embryo without compromising their viability.

For the first isoflavone elicitation experiment, three dilutions of concentration 1000, 100 and 10 µg·mL^−1^ diluted in 10% DMSO were prepared from the separated aliquot of the dry extract obtained by liquid–liquid separation, and 10 µL of each dilution was inoculated into the cut of the seeds embryo. For the second isoflavone elicitation experiment, 10 µL of fractions F1 and F2 at 100 µg·mL^−1^, obtained from the first VLC, was inoculated. For the third isoflavone elicitation experiment, 10 µL of F1.1 at 100 µg·mL^−1^, obtained from the second VLC, was inoculated. For the fourth isoflavone elicitation experiment, 10 µL of Fp at concentrations 100, 10, 1 and 0.1 µg·mL^−1^ was inoculated. This entire procedure was carried out under sterile conditions. Once the treatments were applied, the plates were kept in darkness for 3 days in a SANYO MLR 350H camera at 27 °C.

The extraction and analysis of isoflavones was performed according to Wang et al. [41] and Lozovaya et al. [42] with some modifications. Seeds were powdered with liquid nitrogen, mixed with 100 mL of 80% HPLC-methods methanol and maintained on an orbital shaker at 145 rpm for 15 h at 40 °C. Samples were then centrifuged at 2890× *g* for 20 min at 20 °C. The obtained supernatant was filtered through a 0.2 µm nitrocellulose membrane, and the methanolic extract was used for analysis by HPLC.

The identification and quantification of isoflavone was carried out on a chromatograph Agilent Technologies 1260 Infinity HPLC system. Chromatography conditions were: UV detection: 262 nm, ZORBAZ 300SB-C18 column (4.6 μm × 250 mm × 5 µm), Gecko 2000 30, 80 °C thermostat that kept the column at 30 °C. The mobile phase consisted of water with 0.1% acetic acid (solvent A), and acetonitrile with 0.1% acetic acid (solvent B), with the following gradient: from 15% to 45% of B during 40 min, rising until 100% B during 1 min and remaining this composition for 9 min, after which it dropped to initial conditions (15% B) for 1 min and held for 9 min to equilibrate the column. The flow was 1.5 mL·min^−1^ and the sample injection volume was 10 μL.

The quantification of the isoflavones (μg·mL^−1^) was performed by interpolation of the relative area given by the detector on a calibration curve built for each isoflavone (*R*^2^ > 0.99). The calibration curves were built with the isoflavones (LC Laboratories): daidzin, genistin and malonyl genistin.

### 2.4. Vacuum Liquid Chromatography (VLC)

The two VLCs performed were carried out in a glass column (20 mm diameter × 350 mm H) filled with 30 g of silica gel 60 (0.063–0.200 mm, Merck) coupled to a vacuum pump with 51 kPa. The starting extract for fractionation was crushed and mixed with silica gel 60 until obtaining a fine powder, which was placed on the top of the silica column and fractionated passing through the column the following mobile phases (from lower to higher polarity): hexane, dichloromethane, ethyl acetate and methanol, for the first VLC, and hexane and dichloromethane for the second VLC. Each organic solvents (400 mL) was passed through the column and concentrated in a rotary evaporator under vacuum at 50 °C.

### 2.5. Thin Layer Chromatography (TLC)

Two thin layer chromatographies, using TLC plates of silica gel on aluminum support 60 F254 (Merk), were performed to qualitatively assess the components present in the control 1 and in the fractions obtained from the first VLC (F1, F3, F3 and F4) and in the fractions obtained from the second VLC (F1.1 and F1.2).

The mobile phase used was a mixture of chloroform/dichloromethane/ethyl acetate/methanol (*v*/*v*/*v*/*v*). Ultraviolet light (254 nm and 366 nm) was used for revealing the TLC plates.

### 2.6. Purification of F1.1

F1.1 fraction, the most fractionated one, was dissolved in 1 mL of chloroform and put on a TLC plate, which was imbibed in a dichloromethane mobile phase. The band that appeared at the top of the plate (common to those bands that had appeared in the first and second TLCs) was removed by scraping the silica gel from the TLC aluminum plate. Scraped silica was mixed with 1 mL of 80% methanol for HPLC methods and centrifuged for 10 min at 6500× *g* at room temperature. Precipitated silica was discarded and supernatant (containing the elicitor pure compound/s) was evaporated. The isolated and pure fraction (Fp) (0.39 mg) was tested in soybean seeds and stored protected from light and humidity to later perform another experiment for checking its capacity to induce systemic resistance in *A. thaliana*.

### 2.7. ISR Experiment

An aliquot of the purest fraction (Fp) was dissolved in DMSO 10% at concentrations 10 and 1 µg·mL^−1^ and used for an ISR experiment in *A. thaliana*. The ISR experiment was carried out as follows:

*A. thaliana* wild type Columbia ecotype 0 seeds (provided by the Nottingham Arabidopsis Stock Centre (NASC)) were germinated in quartz sand and two-week-old seedlings were then individually transplanted to 100 mL pots filled with peat/sand mixture (12/5) (60 g per pot). Forty-eight plants per treatment were used; plants were arranged in three replicates, with sixteen repetitions each. Plants were watered with 5 mL of tap water once a week and with 5 mL of half-strength Hoagland solution per plant once a week. Plants were inoculated by soil drench with 50 µL of each treatment: Fp diluted in 10% DMSO at a concentration of 10 µg·mL^−1^ and at 1 µg·mL^−1^, in the first and second weeks after transplant. Control plants were-mock inoculated by soil drench with 50 µL of control 1. Another positive control was added, in which thirty-six plants (three replicates of 12 plants each) were inoculated by soil drench, in the first and second weeks after transplant, with 1 mL of a 10^9^ ucf.mL^−1^
*P. fluorescens* N 21.4 suspension. Four days after the second inoculation, plants were pathogen challenged with the pathogen *Pseudomonas syringae* pv. tomato DC3000. One day before pathogen challenge, plants were maintained with 99% relative humidity to ensure stomata opening in order to allow disease progress. *P. syringae* pv. tomato DC3000 grown for 24 h was centrifuged (10 min at 2890× *g*) and cells were suspended in 10 mM MgSO_4_ to achieve 10^8^ cfu.mL^−1^. Inoculation was done by spraying the entirety of each plant with 250 mL. Plants were incubated in a culture chamber (Sanyo MLR-350H) with an 8 h light (350 μE s^−1^·m^−2^ at 24 °C) and 16 h dark period (20 °C) at 70% relative humidity. All the leaves of twelve plants (four per replicate) of each treatment and of control 1 were harvested at 6, 12 and 24 h after pathogen challenge (hapc), powdered in liquid nitrogen and stored at −80 °C. These plant samples were used for gene expression analysis by qPCR. The 36 remaining plants (12 per replicate) of each treatment and of control 1, and the 36 plants of positive control, were used to record disease severity 72 h after pathogen inoculation as the number of leaves with disease symptoms relative to the total number of leaves. Results were relativized using the disease severity of leaves inoculated with the control 1 extract as 0% protection.

### 2.8. RT-qPCR Experiment

Total RNA was isolated from each replicate with PureLink RNA Micro Kit (Invitrogen, Waltham, MA, USA), DNAase treatment included. RNA purity was confirmed using NanodropTM. A retrotranscription followed by RT-qPCR was performed.

The retrotranscription was performed using iScript tm cDNA Synthesis Kit (Bio-Rad, Hercules, CA, USA). All retrotranscriptions were carried out using a GeneAmp PCR System 2700 (Applied Biosystems): 5 min 25 °C, 30 min 42 °C, 5 min 85 °C, and hold at 4 °C. Amplification was carried out with a MiniOpticon Real Time PCR System (Bio-Rad): 3 min at 95 °C and then 39 cycles consisting of 15 s at 95 °C, 30 s at 55 °C and 30 s at 72 °C, followed by melting curve to check results. To describe the expression obtained in the analysis, cycle threshold (Ct) was used. Standard curves were calculated for each gene, and the efficiency values ranged between 90 and 110%. Results for gene expression were expressed as differential expression by the 2^−ΔΔCt^ method. *Sand* gene (AT2G28390) was used as reference gen [43]. Gene primers used are shown in Table 2.

### 2.9. Tentative Identification and Characterization by UHPLC/ESI–QTOF–MS

#### 2.9.1. Sample Preparation

The remaining content (0.19 mg) of the purest fraction (Fp) was dissolved in 100 µL of methanol, LC-MS grade, and vortex mixed for 3 min. The mixture was then centrifuged at 10,000× *g* for 5 min and supernatant was collected for direct analysis.

#### 2.9.2. UHPLC-MS Analysis

Samples were analyzed on a 1290 Infinity series UHPLC system coupled through an electrospray ionization source (ESI) with Jet Stream technology to a 6545 iFunnel QTOF/MS system (Agilent Technologies, Waldbronn, Germany). For the separation, a volume of 2 μL was injected in a reversed-phase column (Zorbax Eclipse XDB-C18 4.6 × 50 mm, 1.8 µm, Agilent Technologies) at 40 °C. The flow rate was 0.5 mL·min^−1^ with a mobile phase consisted of solvent A: 0.1% formic acid, and solvent B: methanol. Gradient elution consisted of 2 % B (0–6 min), 2–50 % B (6–10 min), 50–95 % B (11–18 min), 95 % B for 2 min (18–20 min), and returned to starting conditions 2 % B in one minute (20–21 min) to finally keep the re-equilibration with a total analysis time of 25 min. Detector was operated in full scan mode (*m*/*z* 50 to 2000), at a scan rate of 1 scan·s^−1^. Accurate mass measurement was assured through an automated calibrator delivery system that continuously introduced a reference solution, containing masses of *m*/*z* 121.0509 (purine) and *m*/*z* 922.0098 (HP-921) in positive ESI mode; whereas *m*/*z* 112.9856 (TFA) and *m*/*z* 922.009798 (HP-921) in negative ESI mode. The capillary voltage was ± 4000 V for positive and negative ionization mode. The source temperature was 225 °C. The nebulizer and gas flow rate were 35 psig and 11 L·min^−1^ respectively, fragmentor voltage to 75 V and a radiofrequency voltage in the octopole (OCT RF Vpp) of 750 V.

All the solvents used were LC-MS grade. Purified water was obtained from Milli-Q Plus™ System from Millipore (Milford, MA, USA). Formic acid was purchased from Aldrich (St. Louis, MO, USA).

For the study, MassHunter Workstation Software LC/MS Data Acquisition version B.07.00 (Agilent Technologies, Santa Clara, CA, USA) was used for control and acquisition of all data obtained with UHPLC/MS-QTOF.

#### 2.9.3. Data Handling

UHPLC-MS data processing was performed by MassHunter Qualitative Analysis (Agilent Technologies, Santa Clara, CA, USA) Software version B.08.00 using “Molecular Feature Extraction (MFE)” to extract potential molecular features (MFs). The MFE algorithm creates a list of possible components that represent the full TOF mass spectral data features, which are the sum of co-eluting ions that are related by charge-state envelope, isotopic distribution and/or the presence of different adducts and dimmers. Several parameters of the algorithm were set for data extraction, applying 2000 counts as limits for the background noise. Moreover, the algorithm was applied to find co-eluting adducts for the same possible compound, selecting +H, +Na, +K, and neutral water loss as possible adducts for positive ionization and -H, +FA, +Cl in negative ionization. Additionally, the “Generate Formula” option in the MassHunter Qualitative Analysis software was used to generate the empirical formula from accurate mass and isotopic pattern distribution to increase the confidence of compound annotation, with a very good score (about 97–99%).

#### 2.9.4. Compound Identification

The tentative identification of compounds was carried out by comparing their retention times and the accurate masses of features (± 5-ppm error) against online databases, as FOODB, MetaCyc, CEU massmediator, and scientific bibliography.

To reach the possible annotation of peak 2, it was necessary to use the MetaCyc and to consult specific bibliography [44].

The identification of compounds corresponding to peaks 3, 4, 6, 7 and 8 was supported by comparison of their accurate masses in the databases for each compound, providing an accuracy error below 5 ppm. Specific bibliography was also consulted for their annotation [45,46,47]. To confirm the annotation of the peaks 4, 6 and 7, a MS/MS analysis was carried out and their final identification was supported using in silico prediction approaches, such as the freely available tool CFM-ID 3.0. The experimental MS/MS spectra were searched and scored against predicted spectra based on similarity.

Regarding peak 5, it was annotated after matching against different databases, based on its monoisotopic mass and molecular formula.

### 2.10. Statistical Analysis

One-way ANOVA with replicates was used to check the statistical differences in all data obtained. Prior to ANOVA analysis, homoscedasticity and normality of the variance was checked with Statgraphics plus 5.1 for Windows, meeting requirements for analysis. When significant differences appeared (*p* < 0.05) a Fisher test was used [48].

## 3. Results

### 3.1. First Elicitation of Isoflavone in the Soybean: Concentration Optimization

To test the capacity of the extract obtained from the metabolic elicitors by liquid–liquid separation to induce the isoflavone secondary metabolism, this extract was inoculated in the embryo cut of the soybean seeds at concentrations of 1000, 100 and 10 µg·mL^−1^, as is explained in Section 2.3. The fractions of 1000 and 100 µg·mL^−1^ significantly elicited isoflavone production over both controls (Figure 2), and the seeds inoculated with 100 µg·mL^−1^ recorded the highest induction. However, the seeds inoculated with 10 µg·mL^−1^ showed an isoflavone synthesis very similar to that of control 1 and control 2 seeds or even less, in the case of malonyl genistin. Differences between the three concentrations were statistically significant.

As there were not statistical differences between both controls in the synthesis of isoflavones, it was assumed that elicitation was due to the components of the metabolic elicitors and not the components present in the nutrient broth. For that reason, the control 1 was not further purified as the metabolic elicitor extract, and for the following elicitation experiments only control 1 was used.

### 3.2. First TLC

To visually assess the components present in the control 1 (C) and in the four fractions (F1, F2, F3 and F4) obtained from the first VLC, a TLC was performed (Figure 3). In the TLC plate, only F1 and F2 showed separation between their components and both showed a common band at the top of the plate. Control 1 extract did not show band separation.

### 3.3. Second Elicitation of Isoflavone Metabolism in the Soybean

According to the results obtained from the previous TLC, only F1 and F2 (of the first VLC) were tested in soybean seeds to check their ability to elicit isoflavone secondary metabolism (Figure 4), as F3 and F4 did not showed component fragmentation. The biosynthesis of the three isoflavones assessed was significantly higher in the seeds that were inoculated with F1 and F2 fractions compared to control 1 seeds. The highest elicitation values were obtained with F1.

When comparing this experiment with the first elicitation experiment, it was seen that the values of isoflavone elicitation obtained with the F1 and F2 fractions were similar to those obtained with the initial extract.

### 3.4. Second TLC

Since the highest elicitation values were obtained with F1 (Figure 4), a second VLC was performed to fragment and purify it. After the VLC, a TLC was made to visually assess the components present in the two fractions obtained (F1.1 and F1.2). F1.1 and F2.1 showed again a common band at the top of the TLC, bands of F1.1 being more intense than those of F2.1. These bands were very similar to those seen in the first TLC of F1 and F2 (Figure 5).

### 3.5. Third Elicitation of Isoflavone Metabolism in the Soybean

As F1.1 fraction showed a similar but more intense band than F2.1 in the second TLC, it was chosen to check its capacity to elicit isoflavone secondary metabolism and it was inoculated in soybean seeds (Figure 6). Seeds inoculated with F1.1 had a significant higher isoflavone concentration than control 1 seeds.

When comparing this experiment with the first and the second elicitation experiments, it was seen that the values of isoflavone elicitation obtained with the F1.1 fraction were 1.2 times greater than those obtained with the F1 and F2 fractions and with the initial extract.

### 3.6. Fourth Elicitation of Isoflavone Metabolism in the Soybean

As F1.1—the fraction obtained from the second VLC—elicited the isoflavone metabolism, it was purified and the resultant fraction (Fp) was tested at concentrations of 100, 10, 1 and 0.1 µg·mL^−1^ in soybean seeds to check its capacity to elicit secondary metabolism of isoflavone (Figure 7).

Seeds inoculated with 100 and 10 µg·mL^−1^ were not able to germinate; they went black (data not shown) and they were not analyzed. However, in the seeds inoculated with 1 and 0.1 µg·mL^−1^, the synthesis of daidzin and especially of genistin significantly increased with respect to control 1. The synthesis of malonyl genistin decreased compared to control 1. No significant differences were seen between both concentrations in any of the three isoflavones.

When comparing the elicitation results obtained in this last experiment with the previous elicitation experiments (first, second and third), it was seen that the increases in the concentrations of daidzine and genistin triggered by Fp fraction (compared to the control 1) were more than double those produced by the F1.1, F1 and F2 fractions and the initial extract.

### 3.7. ISR Experiment

As the purest fraction (Fp) elicited the isoflavone metabolism in soybean seeds, it was then inoculated in *A. thaliana* seedlings at concentrations of 10 and 1 µg·mL^−1^ to carry out an ISR experiment with the objective of checking its ability to protect plants against *P. syringae* pv. tomato DC 3000 infection. Results of protection against infection are shown in Figure 8.

Protection rates against infection were between 40% and 50% for the Fp fraction and 70% for the live strain (used as positive control). No significant differences between the two concentrations of the Fp fraction were observed.

### 3.8. RT-qPCR Experiment

After the ISR experiment in *A. thaliana*, the differential expressions of marker genes of the salicylic acid (SA) and jasmonic/ethylene (JA/ET) signal transduction pathways were analyzed by qPCR (Table 3). The studied marker genes of the SA signaling pathway were *PR1*, *NPR1* and *ISC*, and those of the JA/ET pathway were *PDF1*, *PR3* and *LOX 2*. All these genes were measured at 6, 12 and 24 hapc.

The differential expression of marker genes of the SA signaling pathway at both concentrations (10 and 1 µg·mL^−1^) decreased from 6 to 12 hapc, except for *PR1*, with 10 µg·mL^−1^, which increased. With 1 µg·mL^−1^, *PR1* did not show significant differential expression, while *NPR1* and *ICS* showed slightly higher expression with 1 µg·mL^−1^ than with 10 µg·mL^−1^ at both sampling moments. None of the genes showed differential expression at 24 hapc.

Marker genes of the JA/ET signaling pathways, at both concentrations, only showed significant differential expression at 6 hapc, except *PR3*, which also showed significant differential expression at 12 hapc with 1 µg·mL^−1^. The expressions of *PDF1* and *PR3* were higher with the concentration of 1 µg·mL^−1^. None of the genes showed differential expression at 24 hapc.

### 3.9. Characterization by UHPLC/ESI–QTOF–MS

The purest fraction (Fp) obtained from the metabolic elicitors of *P. fluorescens* N 21.4 was analyzed by UHPLC/ESI–QTOF–MS, as described in Material and Methods, leading to the characterization of eight peaks (corresponding to eight compounds). Figure 9 shows the extracted ion chromatograms (EICs) provided by the analysis of the extract in the positive ionization mode, which has proved to be more efficient and sensitive than in negative mode for compound characterization, and located within the chromatographic retention interval 14–17.5 min. The tentatively identified compounds, classified by families, and the main parameters that support their annotation, are listed in Table 4.

In the research proposed, eight compounds (Figure 9) were detected in the extract of the purest fraction—among them, one alkaloid (peak 2), two amino lipids (peaks 3 and 8), a terpenoid (peak 5) and three aryl alkylamines (peaks 4, 6 and 7).

According to its monoisotopic mass molecular formula and by consulting a specific database and bibliography [44], the annotation Ambiguine P (a cycloheptane-containing member of the hapalindole alkaloid) could be possible for the compound corresponding to peak 2.

The family that encompasses amino derivatives was remarkable in the extract due to the number of identified compounds (compounds 3, 8, 4, 6 and 7 of the Table 4). The most important compounds in this family were those related to long chain dialkyl, monalkyl and aryl alkylamines, saturated and unsaturated. In this study, peak 3, was annotated as Sphinganine C17 and peak **8** was annotated as *N*-Methyl-*N*-stearylamine (1-nonadecanamine). The peaks 4, 6 and 7, were annotated as arylalkylamines and their identification was supported by their MS/MS spectra (see Appendix A).

Regarding peak 5, (see Appendix A for experimental MS/MS spectra), and considering different databases based on its monoisotopic mass and molecular formula, two terpenoid annotations could be considered to be related: the sesquiterpenoid 4-*O*-methylmelleolide (alkyl resorcinol ester derivative) [49], and the diterpenoid 2-Acetoxy-3-deacetoxycaesaldekarin E (neocaesalpin AH) [50].

## 4. Discussion

The pressure on the demand for agriculture practices that improve yield and quality food, linked to increasing concerns about sustainability, has led to the emerge of studies about new substances that substitute environmentally harmful or health risky phytosanitaries for more ecofriendly and effective plant inoculants able to enhance plants’ immune systems and plants’ resistance to pests and abiotic stress. This current development of biotechnological alternatives in the agronomic field has been concentrated on the use of more efficient tools with low biological and environmental repercussions [23]. One of these reliable and non-polluting tools is the use of inductors or elicitors from beneficial microorganisms [13]. Therefore, the present work was focused on isolating, purifying and identifying the compound or set of elicitor compounds extracted from the metabolism of the beneficial rhizobacterium *P. fluorescens* N 21.4, which were able to elicit isoflavone secondary metabolism in soybean seeds and to induce systemic resistance in *A. thaliana*.

Our results have shown that metabolic elicitor fractions extracted from the strain *P. fluorescens* N 21.4 were able to enhance the synthesis of isoflavones in soybean seeds between 1.2 to 3.2 times more than controls, demonstrating its potential to elicit secondary defense metabolism (Figure 4, Figure 6 and Figure 7). This capacity of elicitation was increased while the purification and concentration of the fractions (from F.1 at 100 µg·mL^−1^ to Fp at 1–0.1 µg·mL^−1^) progressed [51].

The technique provided in the present work to verify the elicitation of secondary metabolism in the plant was an effective, simple and very fast technique, since it allowed checking the eliciting capacity of the fractions inoculated in soybean seeds in less than 90 h. This fast and effective system would allow one to carry out rapid screenings to search for new eliciting compounds in future research.

On the other hand, the results obtained in the ISR experiment in *A. thaliana* against the pathogen *P. syringae* pv. tomato DC 3000 supported the statement that the metabolic elicitors of the strain N 21.4 have great potential to increase plant resistance, since protection rates between 40 to 50% were seen (Figure 8). The ISR experiment also revealed a simultaneous activation of both SA and JA/ET signaling pathways, since high levels of *PR1*, *NPR1* and *ICS* (SA marker genes), and *PDF1*, *LOX 2* and *PR3* (JA/ET marker genes) were seen. Hence, it has been again demonstrated that these two pathways are not necessarily antagonistic, as previously indicated by some authors [52,53]. The importance of high concentrations of SA and JA to trigger defensive responses mediated by both hormones is nowadays widely accepted [3,54]. Furthermore, this experiment showed a very rapid ISR response, having high values of differential gene expression at 6 hapc. Despite not having seen big differences between both concentrations of inoculation (1 and 10 µg·mL^−1^), a trend of greater differential expression and higher protective rates against pathogen infection were seen when plants were inoculated with 1 µg·mL^−1^. The genes that showed the greatest differences of expression between the two concentrations were *ICS* and *PR3* genes, which were doubly expressed in the plants inoculated with 1 µg·mL^−1^. From a commercial point of view, having a noticeable effect at such a low concentration is very interesting when the intention is making effective, but at the same time affordable plant inoculants.

After the exhaustive analysis by UHPLC/ESI–QTOF–MS to identify and characterize the compounds responsible for the elicitation, eight peaks corresponding to eight different compounds were detected (Figure 9). The groups of compounds found corresponded with an alkaloid, two amino lipids, a terpenoid and three arylalkylamines. An unknown compound was also detected, but it was impossible to elucidate its identity since it was not found described in the literature nor in databases.

A tentative annotation was found for compound corresponding to peak 2, the alkaloid nature one, after consulting the MetaCyc database and specific bibliography [45]. This compound could be described as Ambiguine P, a cycloheptane-containing member of the hapalindole alkaloid. Hapalindole-type natural products are structurally diverse terpenoid indole alkaloids that, to date, have only been described as secondary metabolites produced by cyanobacteria. They have a wide range of biological activities, including insecticidal [55], antibacterial and antifungal [56,57,58]. Literature does mention other alkaloid compounds produced by *P. fluorescens*, such as pyrrolnitrin or hydropyridine-type compounds that also show antibiotic and antifungal activity [28,59]. However, hapalindole-type natural products have never been described in *P. fluorescens*; hence, this result would require further investigation to be more conclusive.

Compounds corresponding to peaks 3 and 8 were seen to be amino lipid compounds. Sphinganine C17 (peak 3) is a type of sphingolipid that the literature describes with a signaling function in plants subjected to biotic and abiotic stress [45,46,47]. In bacteria, sphingolipids are specific membrane lipids each with a monounsaturated long-chain, the sphingosine, and their biological role has not been fully understood yet. Few bacteria have been described able to synthesize sphingolipids and some of these are cytophaga-flavobacterium-bacteroidetes group bacteria, α-proteobacteria and δ-proteobacteria (*Bdellovibrio bacterivorous*, *Cystobacter fuscus*, *Myxococcus stipitatus*, *Sorangium cellulosum* and *Myxococcus xanthus* [47]. However, it has been seen that lipopolysaccharide of Gram-negative bacteria shows structural and functional resemblance to sphingolipids of the above-cited bacteria and even to sphingolipids produced by eukaryotes. That is why in the work of Heaver et al. [60] it was proposed that as sphingolipid produced by bacteria are very similar to those produced by their eukaryotic hosts, they could influence in their hosts immune responses.

The role that sphingolipids play in plant-microorganism interactions and as bioactive elicitors that initiate defensive responses in plants has begun to be studied [47]. However, Giorni et al. [61] and Dall’Asta et al. [62] have already established a possible relationship between sphingolipids produced by various maize hybrids with *Fusarium verticillioides* infection.

In contrast to eukaryotes, where sphingolipids functions have been extensively studied, very little is known about sphingolipids in bacteria and their functions in the plant–pathogen relationship. Some of the sphingolipids-producing bacteria have been found to be abundant in the phyllosphere of plants [63], which hints that sphingolipid production by bacteria may be relevant for them.

This study was the first synthesis of sphingolipids in the genus *Pseudomonas*, and its relationship with the triggering of secondary metabolism is cited. However, it is clear that more research is needed on this unknown subject.

The other amino lipid nature compound, the one corresponding to peak 8, was tentatively identified as *N*-methyl-*N*-stearylamine (1-nonadecanamine). This compound has been reported in some works for its antimicrobial potential, being more commonly found in plant species [64,65,66]. It has been also found as a secondary metabolite of *Brevibacterium casei* [67] and *Saccharomyces cerevisiae*, with antibiotic activity against *Proteus mirabilis* [68]. Compound 8 has also been described in other bacterial and plants extracts as a structural constituent of biological membranes, and the results were consistent with those of previous literature reports [69].

The amino derivatives corresponding to arylalkylamines (peaks 4, 6 and 7) have been generally described as dialkyl and monalkyl amines, saturated and unsaturated. Further investigations will be necessary to determine whether such alkylamines may be considered functional analogs of sphingosine and which may be primarily expressed among the pathways associated with sphingolipid metabolism.

For the last identified compound, the most abundant one (peak 5), two possible annotations within the group of terpenoids were found: the sesquiterpenoid 4-*O*-methylmelleolide (alkyl resorcinol ester derivative) [49] and the diterpenoid 2-acetoxy-3-deacetoxycaesaldekarin E (neocaesalpin AH) [50].

Terpenoids are generally considered as secondary metabolites produced by plant or fungi, but recent sequencing of the bacterial genome and bioinformatics analyses of bacterial proteins have revealed the presence of these metabolites in bacteria [70,71]. In bacteria, terpenoids can be found in the form of essential oils or aromatic constituents and some have antibiotic and antifungal activities [72,73].

Despite terpene biosynthetic pathways in bacteria being considered ubiquitous, few bacterial terpenes have been identified, and their biosynthesis is still poorly understood [74]. The antiSMASH tool lists more than 4000 bacterial terpene biosynthetic gene clusters [75], but only 127 have been identified and deposited in the MIBiG database (repository of characterized biosynthetic gene clusters) [76], so far. However, literature has revealed the production of 2,5-dialkylresorcinol compounds that exhibit antifungal and antibacterial activities in specific strains of *Pseudomonas* spp. [77,78]. These are compounds very similar to the compound proposed by our first annotation option (alkyl resorcinol ester derivative).

The induction of systemic resistance in *A. thaliana* and the elicitation of the secondary isoflavone metabolism in soybean seeds may have been due to the effect of the sesquiterpenoid compound present in the extract of the purest fraction of the metabolic elicitors of *P. fluorescens* N 21.4—it being the majority in the fraction—or due to a synergy between all the compounds that have been identified. Nevertheless, it is clear that deeper research will be necessary in the future to carry out more specific studies with the identified compounds in order to better characterize them and their effects in the plants.

By comparing the fractions extracted from the metabolic elicitors with the control 1, which was extracted in the same way as the fractions, it has also been possible to affirm that the inducing and protective effects were due to the compounds present in the metabolic elicitors of the bacterium and not to any compound present in the nutrient broth. Furthermore, it has been seen that the beef extract used in the nutrient broth is a concentrate of water-soluble compounds, which remained in the aqueous phase, which was discarded in the initial liquid-liquid separation.

In our work we tested the capacity to induce systemic resistance and the capacity to elicit secondary defensive metabolism in plants, but in view of our promising results and having some compounds with putative antibiotic and/or antifungal activity, further research will be performed to test their potential as antibiotics or antifungals against common pathogens present in agricultural systems. Furthermore, we will study the possibility of including all or some of these compounds, derivatives of the secondary metabolism of the *P. fluorescens* N 21.4, as commercial plant inoculants.

## 5. Conclusions

The results of the present study demonstrated that the fractions obtained by VLC from the metabolic elicitors of *P. fluorescens* N 21.4 induced systemic resistance in *A. thaliana* seedlings against the pathogen *P. syringae* pv. tomato DC3000, being able to trigger the two signaling pathways of the defensive response (SA and JA/ET) very quickly and at very low concentrations. These fractions also boosted the secondary defensive metabolism of isoflavones in soybean seeds.

Taking into account our results and those provided in previously published works, it can be concluded that the metabolic elicitors of *P. fluorescens* N 21.4 could be used to create new plant inoculants to be introduced in agricultural practices, minimizing the scope of chemical control, and thus advancing the development of ecofriendly agricultural tools. The purified and identified compounds of the metabolic elicitor fraction (mixed or individually), could result in commercial products of biological origin being applied to crops in the near future, since many of them have putative antibiotic and/or antifungal potential.

Furthermore, the elicitor screening system carried out in soybean seeds to specifically study the metabolic elicitors’ effect on isoflavone metabolism, is a doubly fast and efficient system that allows one to verify the elicitation of secondary metabolism in the plant in less than 90 h, which will be very useful for future elicitor compound screenings.

In this work, in *Pseudomonas fluorescens*, we described the synthesis of certain compounds that had not been described in the literature to date, such as sphingolipids and hapalindole-type natural products.

## Figures and Tables

**Figure 1 plants-09-01020-f001:**
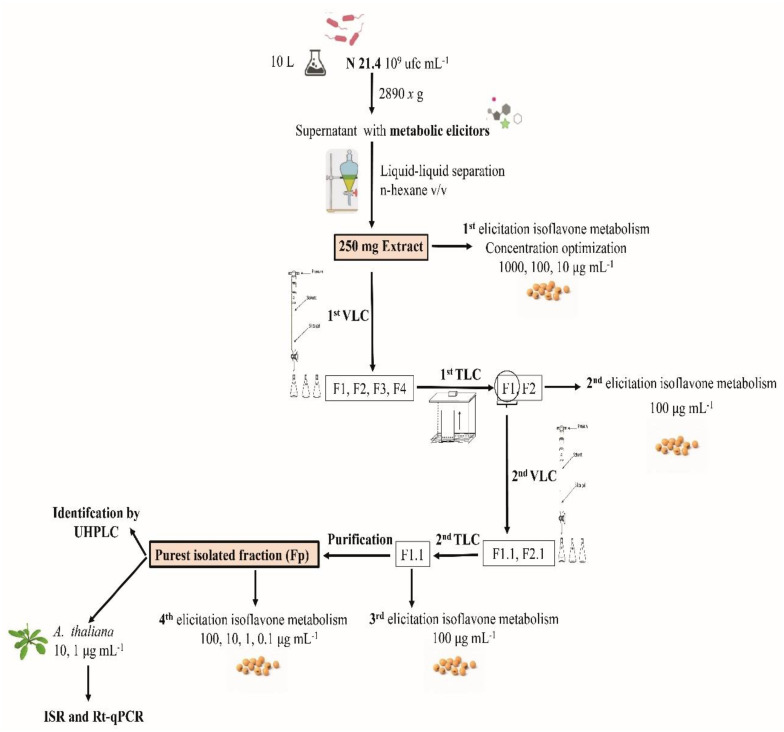
Representative scheme of the sequential extraction, fractionation and purification of the metabolic elicitors of *P. fluorescens* N 21.4. Shown are the growth of the bacterium in nutrient broth, the liquid–liquid phase separation with *n*-hexane, the process of fractionation and purification by two sequential vacuum liquid cromatographies, the four elicitation experiments of isoflavone metabolism, the ISR experiment in *A. thaliana* and the final compound identification by UHPLC.

**Figure 2 plants-09-01020-f002:**
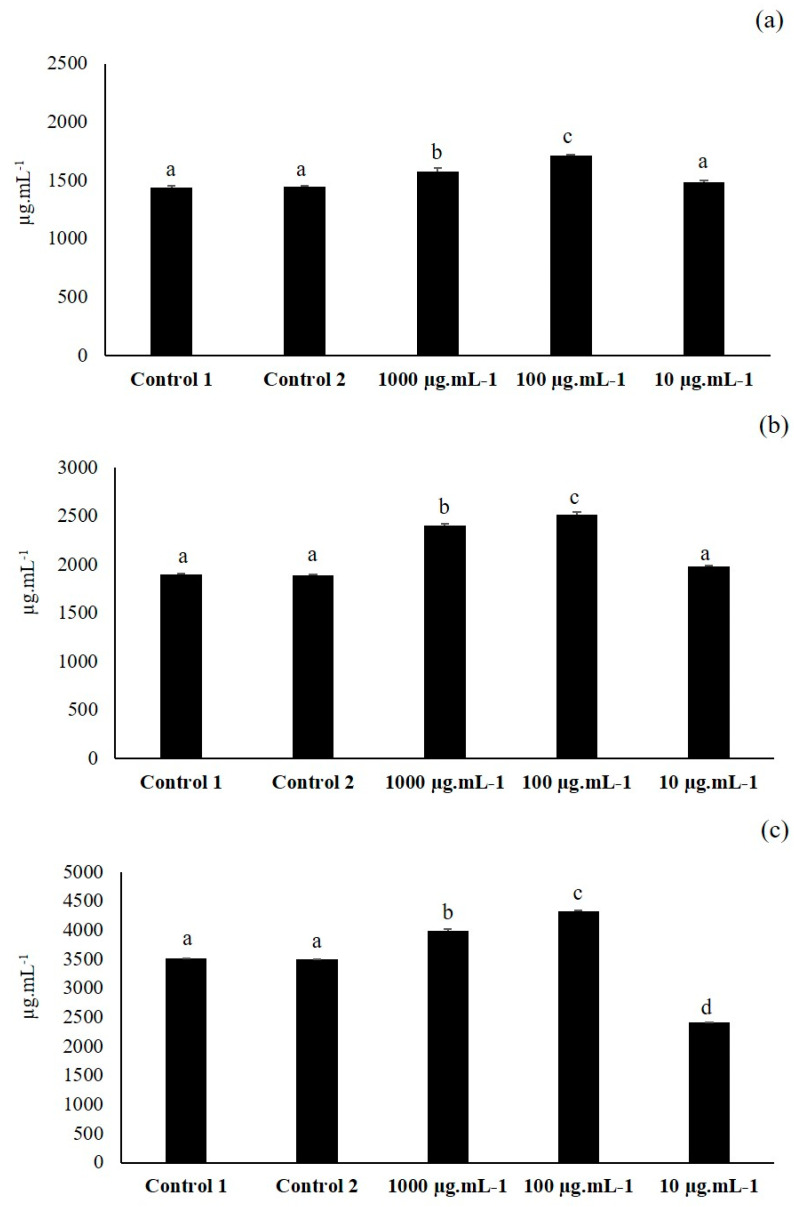
Daidzin (**a**), genistin (**b**) and malonyl genistin (**c**) production in the seeds inoculated with the extract obtained from the metabolic elicitors at 1000, 100 and 10 µg·mL^−1^ and in control 1 (extract obtained from the liquid–liquid phase separation from the culture broth without bacteria) and control 2 (non-inoculated soybean seeds). The amount of isoflavones is expressed in µg·mL^−1^ (*n* = 30 soybean seeds × 3 replicates). Different letters show significant statistical differences between treatments in each isoflavone (*p* < 0.05). Error bars correspond to standard deviation (SD).

**Figure 3 plants-09-01020-f003:**
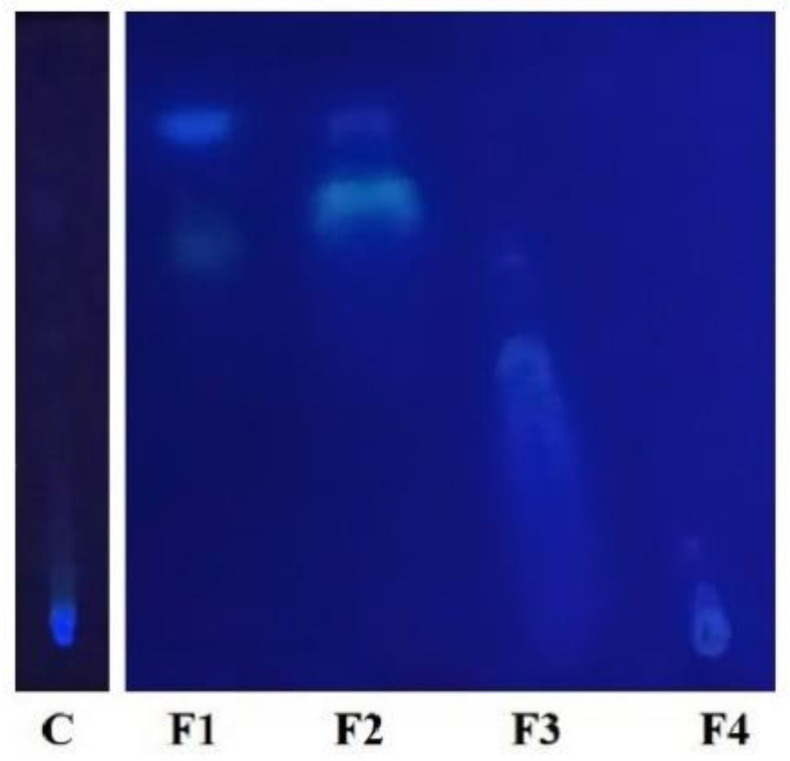
First TLC made with the control 1 (C) and the four fractions obtained from the fist VLC. F1 is the fraction obtained with hexane, F2 with dichloromethane, F3 with ethyl acetate and F4 with methanol.

**Figure 4 plants-09-01020-f004:**
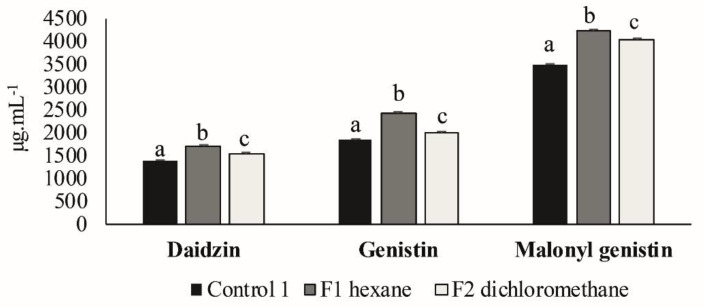
Daidzin, genistin and malonyl genistin production in the seeds inoculated with F1 and F2 obtained from the first VLC and in control 1 seeds. The amount of isoflavones is expressed in µg·mL^−1^ (*n* = 30 soybean seeds × 3 replicates). Different letters show significant statistical differences between treatments in each isoflavone (*p* < 0.05). Error bars correspond to standard deviation (SD).

**Figure 5 plants-09-01020-f005:**
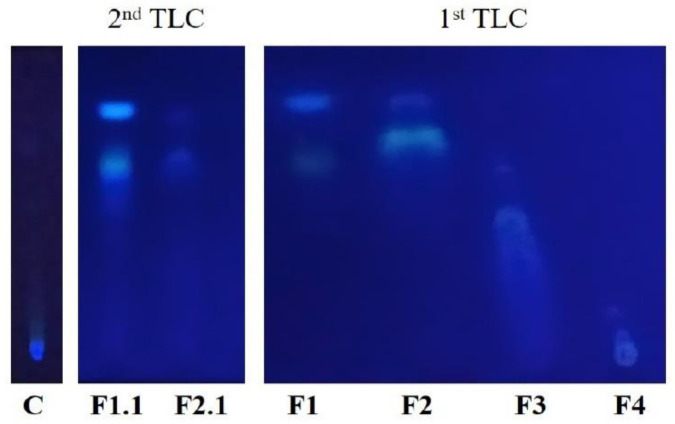
Comparison between the first and the second TLC. Second TLC was made with the two fractions obtained from the second VLC. F1.1 is the fraction obtained with hexane and F2.1 with dichloromethane.

**Figure 6 plants-09-01020-f006:**
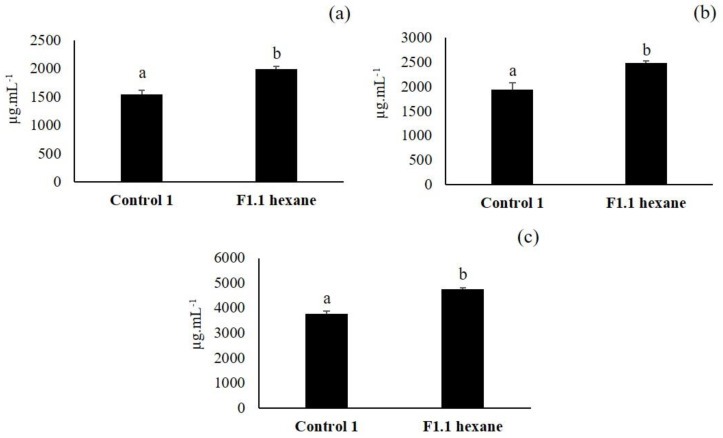
Daidzin (**a**), genistin (**b**) and malonyl genistin (**c**) production in the seeds inoculated with F1.1 obtained from the second VLC and in control 1 seeds. The amount of isoflavones is expressed in µg·mL^−1^ (*n* = 30 soybean seeds × 3 replicates). Different letters show significant statistical differences between treatments in each isoflavone (*p* < 0.05). Error bars correspond to standard deviation (SD).

**Figure 7 plants-09-01020-f007:**
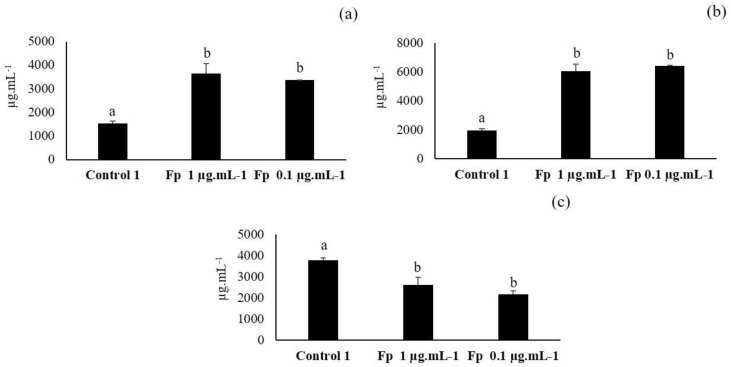
Daidzin (**a**), genistin (**b**) and malonyl genistin (**c**) production in the seeds inoculated with the purest fraction (Fp) obtained from F1.1 and in control 1 seeds. The amount of isoflavones is expressed in µg·mL^−1^ (*n* = 30 soybean seeds × 3 replicates). Different letters show significant statistical differences between treatments in each isoflavone (*p* < 0.05). Error bars correspond to standard deviation (SD).

**Figure 8 plants-09-01020-f008:**
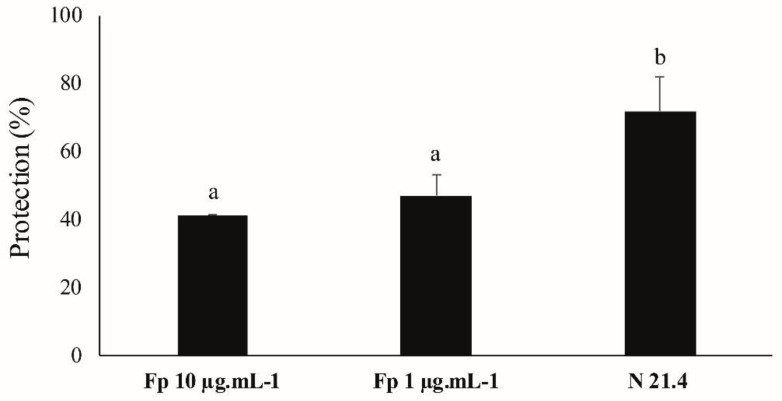
Protection (%) against *Pseudomonas syringae* pv. tomato DC3000 in *Arabidopsis thaliana* seedlings elicited with the purest fraction (Fp) at 10 and 1 µg·mL^−1^ and with the live strain *P. fluorescens* N 21.4. The percentage was calculated based on the number of leaves with disease symptoms to the total of leaves (*n* = 12 seedlings per replicate). Data were relativized to control 1, which was considered as 0% protection. Different letters show significant statistical differences between treatments (*p* < 0.05). Error bars correspond to standard deviation (SD).

**Figure 9 plants-09-01020-f009:**
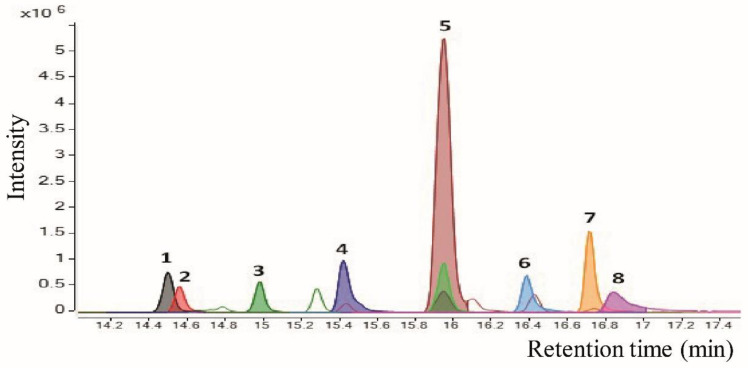
Overlay of the extracted ion chromatograms (EICs) for compounds present in the extract of the purest fraction. The enlargements of a part of the chromatogram are shown from 14.0 to 17.5 min.

**Table 1 plants-09-01020-t001:** Experimental design and treatments used.

Method Used	Treatment	Characteristics	Use
Growth in nutrient broth	*P. fluorescens* N 21.4	10 L 10^9^ ufc.mL^−^^1^	Liquid-Liquid separation ISR as positive control
Liquid-Liquid separation (bacterial culture)	Dry extract	250 mg Aliquots dissolved in 10% DMSO	1st Isoflavone elicitation: Concentration optimization (1000,100,10 µg·mL^−1^) 1st VLC
Liquid-Liquid separation (nutrient broth)	Control 1	Dissolved in 10% DMSO	1st,2nd,3rd and 4th isoflavone elicitation 1st and 2nd TLCs ISR as 0% of protection
Seed embryo cut	Control 2	Not inoculated soybean seeds	1st Isoflavone elicitation
1st VLC with dry extract	F1 F2 F3 F4	Hexane Dichloromethane Ethyl acetate Methanol	1st TLC 1st TLC 1st TLC 1st TLC
1st TLC with F1, F2, F3 and F4	F1 F2	100 µg·mL^−^^1^ 100 µg·mL^−^^1^	2nd Isoflavone elicitation
2nd VLC with F1	F1.1 F1.2	Hexane Dichloromethane	2nd TLC 2nd TLC
2nd TLC with F1.1 and F1.2	F1.1	100 µg·mL^−^^1^	3rd Isoflavone elicitation
Purification of F1.1	Fp (Purest fraction)	1, 0.1 µg·mL^−^^1^	4th Isoflavone elicitation
ISR in *A. thaliana*	Fp	10, 1 µg·mL^−^^1^	Pathogen protection (%) qPCR (SA and JA/ET marker genes)
UHPLC/ESI–QTOF–MS	Fp	0.19 mg in 100 mL methanol LC-MS grade	Tentative compound identification

**Table 2 plants-09-01020-t002:** Forward and reverse primers used in qPCR analysis.

	Forward Primer	Reverse Primer
*AtNPR1*	5′-TATTGTCAARTCTRATGTAGAT	5′-TATTGTCAARTCTRATGTAGAT
*AtPR1*	5′-AGTTGTTTGGAGAAAGTCAG	5′-GTTCACATAATTCCCACGA
*AtICS*	5′-GCAAGAATCATGTTCCTACC	5′AATTATCCTGCTGTTACGAG
*AtPDF1*	5′-TTGTTCTCTTTGCTGCTTTCGA	5′-TTGGCTTCTCGCACAACTTCT
*AtLOX2*	5′-ACTTGCTCGTCCGGTAATTGG	5′-GTACGGCCTTGCCTGTGAATG
*AtPR3*	5′-AAATCAACCTAGCAGGCCACT	5′-GAGGGAGAGGAACACCTTGACT
*Sand*	5′-CTGTCTTCTCATCTCTTGTC	5′-TCTTGCAATATGGTTCCTG

**Table 3 plants-09-01020-t003:** Differential gene expression (*A. thaliana* seedlings inoculated with the purest fraction at a concentration of 10 and 1 µg·mL^−1^ vs. control 1) at 6 (*n* = 12), 12 (*n* = 12) and 24 (*n* = 12) hapc.

		10 µg·mL^−1^			1 µg·mL^−1^		
		6 hapc	12 hapc	24 hapc	6 hapc	12 hapc	24 hapc
SA	*PR1*	1.2 ± 0.02 *	5.35 ± 0.06 *	0	0.83 ± 0	0	0
	*NPR1*	2.44 ± 0.21 *	1.1 ± 0.05	0.73 ± 0	2.65 ± 0.13 *	1.2 ± 0.03 *	0
	*ICS*	1.66 ± 0.12 *	0.66 ± 0.01 *	0	2.02 ± 0.03 *	1.47 ± 0.03 *	0
JA/ET	*PDF1*	1.23 ± 0.01 *	0	0	2.36 ± 0.02 *	0	0
	*LOX2*	1.79 ± 0.06 *	0.73 ± 0.03	0	0.9 ± 0.01	0	0
	*PR3*	1.7 ± 0 *	0.72 ± 0	0	3.4 ± 0.06 *	1.31 ± 0.1 *	0

Asterisks represent statistically significant differences (*p* < 0.05) with respect to control 1 (differential expression of 1) within each sampling time.

**Table 4 plants-09-01020-t004:** Retention times, mass spectral data and characterizations of the detected compounds in the UHPLC/ESI–QTOF–MS analysis in positive and negative ionization modes.

nº	Tentative Annotation *^a^*	Rt (min)	Molecular Formula	Monoisotopic Mass	*m*/*z* Experimental	Fragments (MS2)
**Unknowns**						
1	Unknown	14.5	C_17_H_14_N_2_S	278.0878	[M + Na]^+^ = 301.0762	-
**Alkaloids**						
2	-	14.6	C_25_H_29_NO	359.2249	[M + H]^+^ = 360.2333	-
**Amino lipids**						
3	Sphinganine C17	14.9	C_17_H_37_NO_2_	287.2824	[M + H]^+^ = 288.2999 [M + Na]^+^ = 310.	-
8	1-Nonadecanamine	16.8	C_19_H_41_N	283.3239	[M + H]^+^ = 284.3320	-
**Terpenoids**						
5	-	15.9	C_24_H_30_O_6_	414.2042	[M + Na]^+^ = 437.1946 [M + H]^+^ = 415.2121 [M + K]^+^ = 453.1680 [2M + Na]^+^ = 851.3989 [M + FA-H]^−^ ^=^ 459.2029	303.1214, 73.0661 338.4838, 325.1891, 277.1818, 141.4137, 104.2879, 90.6591
**Arylalkylamines**						
4	*N*-benzyl-1-tetradecanamine	15.4	C_21_H_37_N	303.2926	[M + H]^+^ = 304.3003	212.2379, 91.0544, 65.0383, 58.0652
6	*N*-benzyl-1-hexadecanamine	16.4	C_23_H_41_N	331.3239	[M + H]^+^ = 332.3320	240.2682, 91.0544, 69.0694, 58.0652
7	*N*-benzyl-1-octadecanamine	17.3	C_25_H_46_N	325.3709	[M + H]^+^ = 360.3622	268.2993, 91.0544, 85.0652, 58.0652

*^a^* Compounds identified compared with data reported in literature and online databases.

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
