# Peer review of "Identifying the Compounds of the Metabolic Elicitors of Pseudomonas fluorescens N 21.4 Responsible for Their Ability to Induce Plant Resistance"

_plants, 2020, doi:10.3390/plants9081020_

Round 1

Reviewer 1 Report

The authors in their manuscript “ Identifying the compounds of the metabolic elicitors of Pseudomonas fluorescens N 21.4 responsible for their ability to induce plant secondary metabolism and systemic resistance” presented a very interesting overall dataset as an outcome from a well-organized experiment. Nevertheless, there are several issues that must be corrected before this manuscript can be accepted for publication.

Firstly, in the M&M section, the authors must include a table where they explain their full experimental design and all their treatments. Apart from Fig. 1, which is a nice presentation of the project, the design is scattered all around the M&M and this is not helpful for the reader. Moreover, as they have two controls, they have to explain what these two stands for again in the results (lines 311-312); while this could be avoided by creating a single table with all the abbreviations included. Additionally, the number of sections in M&M should be reduced, no need for them to be so many. Finally, there is no need to have the number of figures (Fig. 2-7) in M&M. We only describe our methods here and these figures are part of the results.  

Part of the results (lines 440-460) should move to M&M. there is no point for all this information to be in the results section. Keep only what explains the graph that you present. Same thing in the discussion: lines 474-500 and 527-533 belong to the introduction (some of them have already been mentioned there).  

Other points

In all graphs, a clear y-axis (solid line ) must be added. Also, in fig. 7 add y-axis titles.

There is no point to have so many decimal zeros in the y-axis.

Whenever you have a white bar always use a solid outline.

In your graphs for the controls use the abbreviations that you’ll add in the new table in M&M

Try to use the same colors in all your graphs

The resolution of the graphs and also of Fig.1 must be increased.

In the legends, everything that is found in the graph needs to be explained (e.g. in fig. 2 the controls are not explained properly.

In the abstract, all species names should be written in italics.

Several parts of the manuscript need linguistic editing.

Reviewer 2 Report

The manuscript title “Identifying the compounds of the metabolic elicitors of Pseudomonas fluorescens N 21.4 responsible for their ability to induce plant secondary metabolism and systemic resistance” contains characterization information of metabolic elicitors of Pseudomonas fluorescens that played an important role in the ISR of a plant. The manuscript contains vital information, and it needs minor revision to improve before publication.

  1. Title is too long, so the author should reduce the length of the title.
  2. Improve the DPI (>300) of fig 1, increase the font size to make figure reader-friendly and elaborate the figure in the caption briefly.
  3. Line no. 104 to 108: This paragraph should move to the introduction as this is not part of methods of the current study, and it is better to mention how the strain maintain in the present study.
  4. Figure 2 presented poorly. Please add the title of X and Y axes and delete “.00” from the Y axes. Increase the font size to make figure reader-friendly. Also, add the no. of replicates in the bar and what error bar representing ( SE? or SD?).
  5. Figure 4: figure needs improvement. Add the bar line to improve the figure intensity and add the no. of replication and what error bar representing ( SE? or SD?).
  6. Figure 6, 7, 8 and 9: Improve the figure quality as per the above comments like DPI, replicates and error bar, Increase font size.
  7. Table 2: please add the control in the table.
  8. The conclusion needs improvement to justify the results.

Reviewer 3 Report

The manuscript describes a very interesting aspect of research improving the knowledge in the elicitors extracted from the beneficial rhizobacterium 15 Pseudomonas fluorescens N 21.4. Authors identified a class of  compounds that could be part of commercial plant inoculants of biological and sustainable origin to be applied in crops, due enhancing plant immune response. The authors check the role of this molecules in plants even if more studies should be conduct to be sure of the role as immuno plant elicitor. I suggest to report more references on the role of the selected molecules in plant-pathogen tinterction to support the hypotesis, such as maize-Fusarium verticillioides and sphingolipids (GIorni et al 2015).
